# A Rare Complication during Vaginal Delivery, Hamman’s Syndrome: A Case Report and Systematic Review of Case Reports

**DOI:** 10.3390/ijerph19084618

**Published:** 2022-04-12

**Authors:** Marco La Verde, Adriano Palmisano, Irene Iavarone, Carlo Ronsini, Domenico Labriola, Stefano Cianci, Ferdinando Schettino, Alfonso Reginelli, Gaetano Riemma, Pasquale De Franciscis

**Affiliations:** 1Obstetrics and Gynecology Unit, Department of Woman, Child and General and Specialized Surgery, Università degli Studi della Campania “Luigi Vanvitelli”, 80138 Naples, Italy; marco.laverde88@gmail.com (M.L.V.); adrianopalmisano@gmail.com (A.P.); ireneiavarone2@gmail.com (I.I.); carlo.ronsini@unicampania.it (C.R.); domenico.labriola@unicampania.it (D.L.); gaetano.riemma7@gmail.com (G.R.); pasquale.defranciscis@unicampania.it (P.D.F.); 2Department of Obstetrics and Gynecology, University of Messina, 98122 Messina, Italy; 3Department of Precision Medicine, Università degli Studi della Campania “Luigi Vanvitelli”, 80138 Naples, Italy; fer.schet@gmail.com (F.S.); alfonso.reginelli@unicampania.it (A.R.)

**Keywords:** uncommon complication, chest discomfort, spontaneous pneumomediastinum, pregnancy, mediastinal emphysema, active labor stages, subcutaneous emphysema, chest tube drainage

## Abstract

Background: Spontaneous pneumomediastinum (SPM) during pregnancy or labor is a rare event. We presented a case report and a systematic review of the literature to provide comprehensive symptoms, treatments, and complications analysis in the pregnant population affected by SPM. Methods: We conducted a comprehensive search of four databases for published papers in all languages from the beginning to 1 September 2021; Results: We included 76 papers with a total of 80 patients. A total of 76% patients were young primiparous, with a median age of 24 ± 5.4 years. The median gestational age was 40 ± 2.4 weeks, with a median duration of labor of 7.4 ± 4.2 h. In 86%, the ethnic origin was not specified. SPM develops in 55% of cases during the second stage of labor. Subcutaneous swelling and subcutaneous emphysema were present in 91.4%. Chest pain and dyspnea were present in 51.4% and 50% of the patients, respectively. We found that 32.9% patients had crepitus, and less common symptoms were dysphonia and tachycardia (14.3% and 14.3%, respectively). Oxygen and bronchodilators were used in 37.7% of the cases. Analgesics or sedatives were administered in 27.1%. Conservative management or the observation was performed in 21.4% and 28.6%, respectively. Antibiotics treatment was offered in 14.3%, whereas invasive procedures such as chest-tube drainage were used in just 5.7% of patients. There were no complications documented in most SPM (70.0%). We found that 16.7% of the SPM developed a pneumothorax and 5% developed a pneumopericardium.; Conclusions: In pregnancy, SPM occurs as subcutaneous swelling or emphysema during the second stage of labor. The treatment is usually conservative, with oxygen and bronchodilators and a low sequela rate. A universal consensus on therapy of spontaneous pneumomediastinum in pregnancy is necessary to reduce the risk of complications.

## 1. Introduction

Pneumomediastinum (PM) is defined as free air in the mediastinal cavity [1]. It is usually the result of a known condition, such as trauma or other chronic pulmonary diseases. Nevertheless, spontaneous pneumomediastinum (SPM), also known as Hamman’s syndrome, occurs as a PM without a determined aetiology, and it can happen after physiologic or pathologic processes [2]. Labor is a physiological process that may induce an SPM, which is frequently misdiagnosed as other clinical emergency conditions [3,4]. SPM has an incidence of 1:44,000 [5,6], while its incidence during labor is approximately 1:100,000 [7]. Concerning its physiopathology, Macklin proposed that alveolar ruptures cause air dissection along bronchovascular sheaths, allowing inspired air into the mediastinum [8,9]. Numerous factors may contribute to increase alveolar pressure, such as a prolonged second stage of labor [10,11,12,13,14,15]. Nevertheless SPM occurs most frequently in young primiparous women who lack predisposing factors [2]. Breathlessness, hoarse voice, vomiting, cough or subcutaneous neck emphysema, and chest discomfort represent the main clinical signs [16]. The SPM diagnosis is clinical; in addition, a CT scan or RX thorax is practiced to exclude other complications as mediastinitis, pneumothorax and heart tamponade [17]. Chest X-ray images may show an extraluminal air presence around the mediastinal structures [18]. Occasionally, in case of occult SPM, mediastinal air can be visualized only with a CT scan [18]. SPM remains a benign pathology, and management is usually conservative [16]. Therefore, a CT scan or RX thorax is performed as part of the follow-up procedure [19]. Analgesics, bed rest, oxygen therapy, bronchodilators, and, on rare occasions, antibiotic treatment are practiced in the hospital setting [2,20,21,22,23]. We presented a case report and a systematic review of the literature to characterize the clinical presentation, treatment, and complications of SPM during pregnancy.

## 2. Materials and Methods

We searched on PubMed, Google Scholar, clinicalTrials.gov, and EMBASE for studies published in all languages from the beginning to 1 September 2021. We adopted MeSH headings, text words, and word variants for “Pneumomediastinum”, “Hamman’s syndrome”. We combined these with terms related to “delivery”, “pregnancy”. We screened titles and abstracts of all citations for potentially relevant papers. Full texts were assessed independently by two authors (A.P. and I.I.) for content, data extraction, and analysis. Pertinent additional manuscripts were recognized from reference lists of reviews and editorials. Records selection observes the PRISMA (Preferred Reporting Items for Systematic Reviews and Meta-Analysis) guidelines [24]. We considered every publication (as a retrospective study, case report or case series) that described pneumomediastinum during labor and delivery. Reports about SPM occurring in the early gestational weeks or in nonpregnant patients were excluded. Data describing the management of symptoms and complications were reported. According to the flow diagram, we deleted 180 articles that were unsuitable to the topic of our review (Figure 1). Among the 152 records remaining, we excluded 19 citations and 12 articles with abstracts not available. Thus, full-text studies assessed for eligibility were 133: 4 review articles, 15 duplicates, 7 non-postpartum pneumomediastinum, and 1 article in press were removed (Figure 1).

## 3. Results

### 3.1. Case Report

In March 2021, our hospital admitted a 23-year-old healthy primigravida in her 41st week of pregnancy. Her prolonged pregnancy was uncomplicated, according to regular prenatal testing. She denied alcohol habits and quit smoking throughout the early stages of pregnancy. Her body mass index was 24 kg/m^2^. Due to hypothyroidism detected during pregnancy, she was taking Levothyroxine 50 mcg/day [25]. No other medical conditions were reported. She reported previous anaphylaxis to penicillin. Given her Bishop score of 2, according with our clinical routine, labor was induced intravaginally with a prostaglandin insert (10 mg controlled release dinoprostone vaginal insert, Propess^®^, Ferring S.p.A, Milan, Italy) [26]. After 12 h, the insert was removed, and the next day the patient went into the delivery room, continuing induction of labor with Oxytocin 5 IU infusion at a rate of 4 mUI/minute. When she entered the delivery room, her cervix was 5 cm dilated, the fetus had a vertex presentation, and his heart rate was eurythmic. The first stage of labor endured for 2½ h and was uneventful. Amniorrhexis occurred near the end of the first stage, and complete dilatation started about an hour later when the patient began pushing vigorously [27]. The fetal well-being was monitored with computerized cardiotocography during labor induction and delivery, and no indicators of fetal distress were noted [28,29,30,31]. However, the expulsive efforts continued to be unsuccessful due to uterine hypokinesia. After 2½ h of active pushing, clinicians chose vaginal operative delivery with Kiwi Omnicup (Clinical Innovations, Muray, UT, USA) [32]. The delivery occurred after two tractions and a mediolateral episiotomy was performed by a gynecologist during the Kiwi application. The placenta was delivered physiologically in 15 min. The liveborn male child’s Apgar score was 8/9 at 1 and 5 min (3430 g). During labor, the pregnant woman reported shortness of breath and middle-side chest pain. Vital signs recorded during the physical examination were blood pressure of 130/80 mmHg, O^2^ saturation of 97% in spontaneous breathing and heart rate of 135 beats per minute. Despite minor exertional dyspnea, the woman’s respiratory condition remained stable. Bilateral chest auscultation revealed no abnormal breath sounds. Due to increased D-dimer readings (6400 ng/L), pulmonary embolism was investigated in the differential diagnosis and a chest angiogram (Chest angio-CT) was performed. Chest angio-CT excluded, as well, pneumothorax, pneumopericardium, and pneumomediastinum. No opacity or bulk filling deficiency in the pulmonary arteries were revealed. However, chest angio-TC revealed the presence of a pneumomediastinum, defined as a collection of air extending from the cardias to the mid-neck area, implying the aetiology of barotrauma associated with Hamman’s syndrome. There was no evidence of parenchymal or pleuro-pericardial lacerations, but the interior layer of the parietal pleura was filled with air (Figure 2).

An inspection of the neck revealed no subcutaneous emphysema. During the initial episode of dyspnea, an arterial blood gas (ABG) study was performed. It revealed a 99% oxygen saturation in spontaneous breathing and slightly alkaline blood gases (pH 7.52), normal pO2 (103 mmHg) and low pCO2 (27.9 mmHg) according to significant tachypnea. After two days, a chest X-ray was performed, which revealed no pneumothorax. The last radiograph (4 days later) revealed that the pneumomediastinum had resolved: the hemidiaphragm remained intact, and no obliteration of costophrenic angles was seen. Acute myocardial infarction was ruled out at the onset of symptoms after repeated cardiac enzyme testing. Clinical outcomes were steady throughout the hospitalization. The woman was actively monitored in the hospital for 6 days without receiving any special treatment. The patient was discharged to her home in good clinical condition, with full eupnea and no additional postpartum problems.

### 3.2. Review

We included 76 papers with a total of 80 patients [7,9,33,34,35,36,37,38,39,40,41,42,43,44,45,46,47,48,49,50,51,52,53,54,55,56,57,58,59,60,61,62,63,64,65,66,67,68,69,70,71,72,73,74,75,76,77,78,79,80,81,82,83,84,85,86,87,88,89,90,91,92,93,94,95,96,97,98,99,100,101,102,103,104]. Patients’ characteristics are summarized in Table 1.

The majority of patients (76%) were young primiparous (median age 24 ± 5.4 years) (Table 1). The median gestational age was of 40 ± 2.4 weeks, with a median duration of labor of 7.4 ± 4.2 h. Many SPMs happened in women with a slightly increased BMI (25.1 ± 1.9 kg/m^2^) (Table 1). There were no common comorbidities or medical treatments among the analyzed patients. The preponderance of publications omitted information about the ethnic origin (86%). SPM occurred in 55% of cases during the second stage of labor and 16% after delivery (Table 2).

Clinically, 91.4% of SPM had subcutaneous swelling and subcutaneous emphysema (Table 3).

Half of the total manifested chest pain and dyspnea (51.4% and 50%), probably due to subcutaneous emphysema narrowing the mediastinal structures (Table 3). A third had crepitus (32.9%), and less common symptoms were dysphonia and tachycardia (14.3 and 14.3%). Additionally, vomiting, coughing, and odynophagia were seen during the SPM in 5.7–4.3 percent of all SPM patients (Table 3). The authors used a large variety of therapies, and 10 authors did not report the treatment used. Oxygen and bronchodilators were utilized in 37.7% of the cases (Table 4).

Analgesics or sedatives were administered in 27.1%. Conservative management or observation was performed in half of the cases (21.4% and 28.6%, respectively, Table 4). Antibiotics treatment was offered in 14.3%, whereas invasive procedures such as chest tube drainage were used in just 5.7% of patients (Table 4). There were no complications documented in the majority of SPM (70.0%); 10 patients (16.7%) developed a pneumothorax and 3 (5%) a pneumopericardium (Table 5). Pharyngeal rupture, hydropneumothorax, takotsubo cardiomyopathy, and esophageal rupture also were uncommon sequelae reported (1.7%, Table 5).

## 4. Discussion

Our review summarizes the main findings of 76 studies addressing SPM during labor. During labor, Simmons first described the SPM presence [9]. Louis Hamman of Johns Hopkins Hospital, described the pneumomediastinum associated with surgical emphysema in 1945 [5]. Hamman’s syndrome appears to be more frequent in nulliparous women and typically manifests during the second stage of labor. However, clinical signs may develop in the end phase of labor or postpartum period, as in our case report. Chest pain, dyspnea, dysphonia, and skin swelling with related crepitations on palpation or movement are common presenting symptoms. When considering the benign and self-limiting Hamman’s syndrome, it is critical to rule out more severe differential diagnoses. The definition of the syndrome itself is not unique, since most authors describe Hamman’s syndrome as an SPM, while others include the presence of subcutaneous emphysema in the definition of the disease. The most common differential diagnoses include spontaneous esophageal rupture, pneumothorax, aortic dissection, myocardial infarction, and pulmonary embolism. The pathophysiology of the SPM is still unknown, and several hypotheses have been advanced. There is probably an increase in alveolar pressure during the second stage of labor, resulting in peripheral alveoli rupture. This phenomenon can be secondary to coughing and bronchospasm associated with asthma and the Valsalva maneuver. An infrequent complication of massive pneumomediastinum is malignant mediastinum (a condition requiring surgical intervention), which manifests in very high mediastinal pressures resulting in shock. A chest X-ray is usually needed to confirm the clinical diagnosis of SPM, although diagnosis was obtained with a chest-CT in our case report. The treatment approach is various due to the lack of a uniform consensus. Oxygen therapy, analgesia, reassurance, and bed rest are commonly used to treat Hamman’s syndrome. Symptoms usually resolve on their own within a few days with no further issues. Recurrence is a rare occurrence. Amine et al. conducted a review, including 34 SPM in pregnancy or labor [3]. In accordance with our review, Amine et al. revealed that most of the SPM were younger primiparas with a term pregnancy. Our review found that the second stage of labor is usually associated with the onset of SPM. This study confirmed that the most common clinical signs were subcutaneous swelling, subcutaneous emphysema, chest pain, and dyspnea. Among the reports included in our review, patients received analgesics, rest, and oxygen until the condition was completely or nearly completely resolved. In half of the cases, symptomatic treatments or observation was administered, with no sequelae reported in the majority of SPM cases. Only a few patients required invasive treatments such as chest tube drainage. The main limitation of our review is the low prevalence of SPM and the lack of data in the literature. Secondly, the totality of the articles are case reports or case series with poor data quality. Another limitation is the wide range of treatments recommended by various hospital protocols, which influences SPM resolution and follow-up. Last, we do not know how pregnancy-induced physiologic changes expose patients to SPM development or how pre-existing risk factors influence SPM onset. The last strength comes from the large number of patients included in the review, representing the most significant number of pregnant women with SPM assessed. Furthermore, we described the papers’ symptoms, treatment, and complications in detail.

## 5. Conclusions

During pregnancy, labor, and delivery, spontaneous pneumomediastinum is rare. According to our literature analysis, most of the cases with Hamman’s syndrome were younger primiparas at terms of pregnancy. The initial signs of the condition are subcutaneous swelling and emphysema, which occur usually during the second stage of labor. In a conservative management setting, the main therapy is oxygen and bronchodilators. More research is required to understand the pathology’s risk factors and reach a consensus on the management of spontaneous pneumomediastinum in pregnancy.

## Figures and Tables

**Figure 1 ijerph-19-04618-f001:**
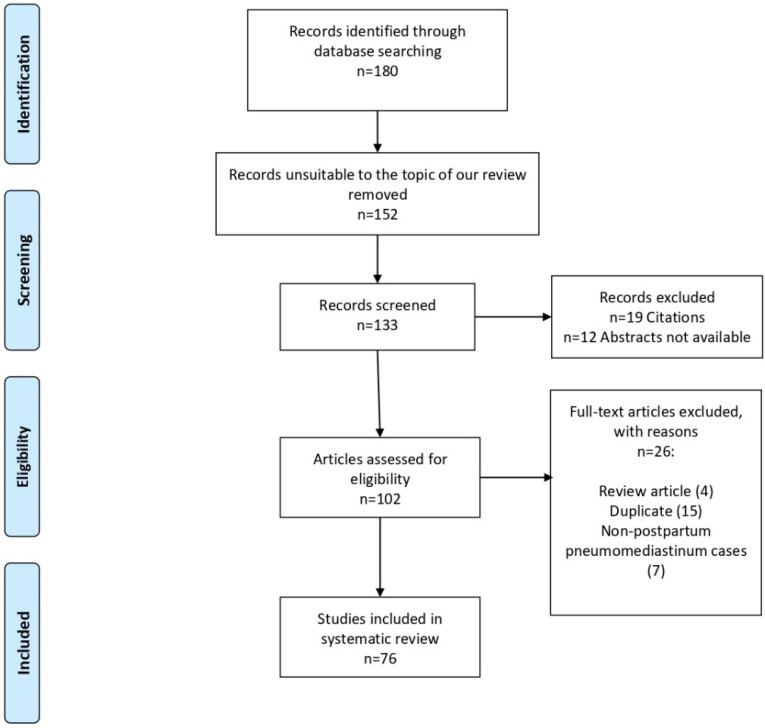
Flow diagram of the study.

**Figure 2 ijerph-19-04618-f002:**
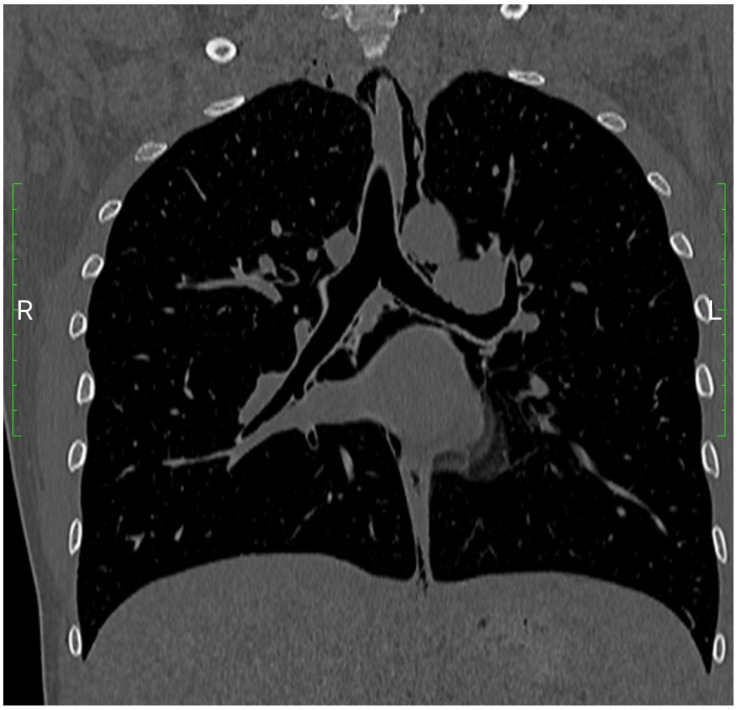
Chest angio-CT.

**Table 1 ijerph-19-04618-t001:** Maternal demographic data.

Age (years), median	24
±SD:	±5.4
Not reported	3
Etnicity, *n*. (%):	
Caucasian	10 (12.5)
African	1 (1.25)
Not reported	69 (86.25)
Parity, *n*. (%):	
1st	61 (76.25)
2nd	4 (5)
3rd	1 (1.25)
4th	1 (1.25)
not reported	13 (16.25)
Hours of Labor, median	7.47
±SD:	±4.2
Not reported	42
Weeks of Gestation, median	40.0
±SD:	±2.4
Not reported	29
Body mass index (BMI, kg/m^2^), median	25.1
±SD:	±1.9
Not reported	31

**Table 2 ijerph-19-04618-t002:** Stage of labor related to pregnancy-associated SPM onset.

Stage of Labor	Number of Cases (%)
2nd stage	44 (55) *
4th stage (after the delivery)	13 (16.25)
1st stage	5 (6.25)
3rd stage	1 (1.25)
Not reported	16 (20)

* *p* < 0.05 relative to other stages.

**Table 3 ijerph-19-04618-t003:** Clinical presentation of pregnancy-associated SPM (Total number = 70, not reported = 10).

Signs and Symptoms	Number of Cases (%)
Swelling & Subcutaneous emphysema	64 (91.4)
Chest pain	36 (51.4)
Dyspnea	35 (50.0)
Crepitus	23 (32.9)
Dysphonia/Hoarse voice	10 (14.3)
Tachycardia	10 (14.3)
Vomiting	4 (5.7)
Cough	3 (4.3)
Odynophagia	3 (4.3)
Neck pain	2 (2.9)
Chest petechiae	2 (2.9)
Hearing loss	1 (1.4)
Suffocation	1 (1.4)
Thoracic back pain	1 (1.4)
Haemoptysis	1 (1.4)

**Table 4 ijerph-19-04618-t004:** Treatment in pregnancy-associated SPM (Total number = 70. Not reported = 10).

Treatment	Number of Cases (%)
Oxygen & bronchodilators	25 (35.7)
Analgesics/Sedatives	19 (27.1)
Observation/None	20 (28.6)
Conservative/Symptomatic treatment (IV Fluids, other)	15 (21.4)
Antibiotics	10 (14.3)
Chest tube drainage	4 (5.7)

**Table 5 ijerph-19-04618-t005:** Complications associated with SPM (Total cases = 60, Not reported = 20).

Treatment	Number of Cases (%)
None	42 (70.0)
Pneumothorax	10 (16.7)
Pneumopericardium	3 (5.0)
Pharyngeal rupture	1 (1.7)
Hydropneumothorax	1 (1.7)
Takotsubo cardiomyopathy	1 (1.7)
Esophageal rupture	1 (1.7)

## Data Availability

The datasets used and/or analyzed during the current study are available from the corresponding author on reasonable request.

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
