# Peer review of "A Rare Complication during Vaginal Delivery, Hamman’s Syndrome: A Case Report and Systematic Review of Case Reports"

_ijerph, 2022, doi:10.3390/ijerph19084618_

Round 1
Reviewer 1 Report
The manuscript is valuable. The presentation is adequate and the content is correct. The text has errors. Authors must correct errors: Line 106 It says: 50g/day It should say: 50 mcg/day Line 128 Dice (6400g/L) It should say: 6400 ng/ml. The authors can delete table 1, it does not provide important data and its reading is difficult (suggestion). The manuscript can be published after errors have been corrected. Congratulations.Author Response
Please see the attachment.

Reviewer 2 Report
This article is case reports of SPM during labor that they experienced and a systematic review of case reports.
I think systematic review from case reports needs new clinical questions to be resolved. But I could not find it in this paper.
The fact that chest pain and subcutaneous emphysema are common symptoms of SPM and that SPM at delivery is common in primipara and 2nd stage of labor has been described in previous study cited by the authors.
Line 40: SPM during the labor, also called Hamman Macklin syndrome, is a condition with an incidence of 1:44.000
- I could not find the description that Hammans syndrome refers to SPM "during labor."
- I think the evidence for the 1 in 44,000 incidence rate was cited from ref 6. But Ref 6(Macia I et al,2007) does not appear to be limited to SPM "during pregnancy". 
- Quotes from Ref 6. 
- Methods: A descriptive, retrospective study of 41 cases -34 men (83%) and 7 women (17%) -treated at our hospital for spontaneous pneumomediastinum from January 1990 through June 2006.
Reviewer 3 Report
Dear Authors,
my small comments:
- In my opinion you should use one name of the disease in all article- in one place is Hamman's syndrome in other Hamman Macklin syndrome.
- Section- 3 Results I am not sure if this order of subsection is correct - in my opinion- Case report might be before Review subsection
- In Case report part- dose of Levothyroxine 50 g/day is very big- probalby should be 50 µg/day.
- Table 3- "not repoted"- should be "not reported" in my opinion; is this statistically significant the 55% cases in 2nd stage?
Round 2
Reviewer 2 Report
Thank you for authors reply.
However, I still think the novelty of this review is weak. Is the novelty of this paper that it confirms that SPM during pregnancy follows the same course as during non-pregnancy or it reconfirm of the fact that it is more common in young primipara and in the second trimester of labor, as described previous papers?
I appriciate that this study represents a massive effort. I believe new perspectives can be found in the authors' painstakingly collected case reports. I believe it is better to re-establish the purpose of conducting the review again. For example, were there any characteristics of patients of SPM in pregnancy, in those BMI (pre pregnancy or before delivery) or past history ,any features or signs of exacerbated cases ?
In some papers(See below), Hamans syndrome is defined as Spontaneous mediastinal emphysema with subcutaneous emphysema.Is this definition correct?
Grapatsas K, et al . Hamman’s syndrome (spontaneous pneumomediastinum presenting as subcutaneous emphysema): A rare case of the emergency department and review of the literature. Respir Medicine Case Reports. 2018;23:63–65. PMCID: PMC5730040
Pain AR et al. Hamman’s syndrome in diabetic ketoacidosis. Endocrinol Diabetes Amp Metabolism Case Reports. 2017;2017(1):17–0135. PMCID: PMC5712834
Kandiah S, Iswariah H, Elgey S. Postpartum Pneumomediastinum and Subcutaneous Emphysema: Two Case Reports. Case Reports Obstetrics Gynecol. 2013;2013:735154. PMCID: PMC3596900
